# Mutual Information Maximization for Robust Plannable Representations

## Abstract

Extending the capabilities of robotics to real-world complex, unstructured environments requires the capability of developing better perception systems while maintaining low sample complexity. When dealing with high-dimensional state spaces, current methods are either model-free, or model-based with reconstruction based objectives. The sample inefficiency of the former constitutes a major barrier for applying them to the real-world. While the latter present low sample complexity, they learn latent spaces that need to reconstruct every single detail of the scene. Real-world environments are unstructured and cluttered with objects. Capturing all the variability on the latent representation harms its applicability to downstream tasks. In this work, we present mutual information maximization for robust plannable representations (MIRO), an information theoretic representational learning objective for model-based reinforcement learning. Our objective optimizes for a latent space that maximizes the mutual information with future observations and emphasizes the relevant aspects of the dynamics, which allows to capture all the information needed for planning. We show that our approach learns a latent representation that in cluttered scenes focuses on the task relevant features, ignoring the irrelevant aspects. At the same time, state-of-the-art methods with reconstruction objectives are unable to learn in such environments.

## 1 Introduction

A fundamental challenge in applying reinforcement learning (RL) to real robotics is the need to define a suitable state space representation. Designing perception systems manually makes it difficult to apply the same reinforcement learning algorithm to a wide range of tasks. Real-world environments are unstructured, cultured, and changing. A suitable representation has to capture features relevant for the task, while ignoring the other elements in the scene. How do we impose inductive biases in the learned representation for real-world robotics?

One could, in principle, apply model-free reinforcement algorithms from raw, low-level observations; however, these tend to be slow, sensitive to hyperparameters, and sample inefficient (Lee et al., 2019). Current work has focused on learning latent spaces for model-based RL constraining them to be feature points (Finn et al., 2016), or learning them with reconstruction objectives (Hafner et al., 2018; Watter et al., 2015; Ha & Schmidhuber, 2018). While feature points objectives provide a good inductive bias, they are too rigid for unstructured tasks and cannot incorporate semantic knowledge of the environment. Learned representations with reconstruction objectives (Kingma & Welling, 2013) force the model to capture all the diversity of the environment in the learned representation in order to accurately reconstruct the observations. As a result, in environments with variability, the relevant features for the task either represent a small component of the latent space or are ignored, both of which prevents deriving a controller from them.

This work tackles the problem of representation learning from an information theoretic point of view: learning a latent space that maximizes the mutual information between the latent and future observations (van den Oord et al., 2018; Hénaff et al., 2019). The mutual information objective induces an inductive bias towards the shared information with future observations, which allows the latent space to discard low-level information and local noise. Hence, by foregoing the reliance on reconstruction objectives, we obtain a latent space that is robust to the presence of variations on the scene if they are unrelated to the task.

The main contribution of our work is a representational learning approach, MIRO (**M**utual **I**nformation Maximization for **RO**bust Plannable Representations), that jointly optimizes for the latent representation and model, and results in a latent space that is robust against disturbances, noisy observations, and achieves performance comparable to state-of-the-art algorithms. We show that previous approaches are brittle even to minor changes in the environment. Finally, we experimentally investigate these effects on the latent variables in our approach and previous work, pinpointing that our learned space is invariant to such changes in the environment. The experimental evaluation is carried out in standard DeepMind Control Suite (Tassa et al., 2018) environments.

## 2 RELATED WORK

**Model-based Reinforcement Learning from Image Observations.** Recent work has shown that model-based reinforcement learning RL is able to achieve the same performance as model-free methods while being substantially more sample efficient (Wang et al., 2019). However, such achievements have been limited to envitonments where a compact state space representation is available, and extending them to raw sensory high-dimensional input spaces poses a major challenge. In such cases, learning a compact and accurate latent space is crucial to relieve the dynamics model from directly modeling the raw sensor input space. Recent work on high-dimensional observations on model-based RL can be categorized in two main classes: 1) video prediction models (Jayaraman et al., 2018; Ebert et al., 2018; Kaiser et al., 2019), and 2) latent space learning with reconstruction objectives (Hafner et al., 2018; Lee et al., 2019; Watter et al., 2015; Wahlström et al., 2015; Zhang et al., 2018; Ha & Schmidhuber, 2018).

Both video prediction models and latent space with reconstruction objectives share the commonality that the latent space is learned using loss on the raw pixel observations. As a result, the latent space model needs to incorporate all information to reconstruct every detail on the observations, which is redundant as the task is usually represented by a small fraction of the scene in real world environments.

**Representation Learning.** Variational autoencoders (VAE) (Kingma & Welling, 2013; Higgins et al.; van den Oord et al., 2017) learn an embedding by optimizing the variational lower bound of the likelihood of data. The learned latent follows a prior distribution, usually Gaussian, which makes sampling and distance meaningful in the latent space. Our method uses a similar architecture to the extension of VAE to sequential data (Chung et al., 2015; Lee et al., 2019; Hafner et al., 2018; Fraccaro et al., 2016). However, instead of a reconstruction objective, we propose to learn a latent space by maximizing the mutual information (MI) between the latent and future observations. The omission of a reconstruction objective relieves the dynamics model from encoding unpredictable variation in the scene, which proves to be more robust when the observation is visually noisy. Our method inspired by the InfoMax principle (Linsker, 1988), learns a latent space by maximizing the mutual information (MI) between observation and latent representation. Despite the difficulty of estimating MI directly, recent work has shown that representations learned by maximizing a noise contrastive estimation lower bound of MI are able to achieve state-of-the-art performance in downstream benchmark tasks in various domains, including image classification, video captioning and natural language processing (van den Oord et al., 2018; Hénaff et al., 2019; Hjelm et al., 2018; Sun et al., 2019; Tian et al., 2019; Bachman et al., 2019). The maximizing the MI has been applied to the context of reinforcement learning, but not in the context of learning robust representations, but to either speed up learning (van den Oord et al., 2018), to achieve hierarchical represenations (Nachum et al., 2018), or model the believe of state space (Gregor et al., 2019; Guo et al., 2018), which use a similar architecture to our method. (Anand et al., 2019) use an MI objective to learn representations for Atari games.

## 3 BACKGROUND

This work tackles the problem of learning a latent space that is suitable for planning from high-dimensional observations in POMDPs. In this work, the observations constitute single images, which are not enough to completely capture the dynamics of the system. For instance, a single image does not contain any information of the velocity or an object might be occluded.

**Partially Observable Markov Decision Process.** A discrete-time finite partially observable Markov decision process (POMDP) $\mathcal{M}$ is defined by the tuple $(\mathcal{S}, \mathcal{A}, T, R, \mathcal{O}, O, \gamma, \rho_0, H)$. Here, $\mathcal{S}$ is the set of states, $\mathcal{A}$ the action space, $T(s_{t+1}|s_t, a_t)$ the transition distribution $(p(s_{t+1}|s_t, a_t))$, $R(s_t$ is the probability of obtaining the reward $r_t$ at the state $s_t$ $(p(r_t|s_t))$, $\mathcal{O}$ is the observation space, $O(o_t|s_t)$ $(p(o_t|s_t))$, $\gamma$ the discount factor, $\rho_0 : \mathcal{S} \to \mathbb{R}_+$ represents the initial state distribution, and $H$ is the horizon of the process. We define the return as the sum of rewards $r_t$ along a trajectory $\tau := (s_0, a_0, ..., s_{H-1}, a_{H-1}, s_H)$. The goal of reinforcement learning is to find a controller $\pi : \mathcal{S} \times \mathcal{A} \to \mathbb{R}^+$ that maximizes the expected return, i.e.: $\max_\pi J(\pi) = \mathbb{E}_{\substack{a_t \sim \pi \\ s_t \sim p}}[\sum_{t=1}^H \gamma^t r_t]$.

**Mutual Information.** The mutual information between two random variables $X$ and $Y$, denoted by $I(X; Y)$ is a reparametrization-invariant measure of dependency. Specifically, it characterizes the Kullback-Leibler divergence between the joint distribution $(X, Y)$ and the product of the marginals $X$ and $Y$: $I(X; Y) = \mathbb{E}_{p(x,y)}\left[\log \frac{p(x|y)}{p(x)}\right] = \mathbb{E}_{p(x,y)}\left[\log \frac{p(x,y)}{p(x)p(y)}\right] = D_{\mathrm{KL}}((X,Y)\|X \otimes Y)$.

Estimating and optimizing the mutual information objective poses a challenging problem. In this work, we use a multi-sample unnormalized lower bound based on noise contrastive estimation, $I_{\mathrm{NCE}}$ van den Oord et al. (2018).

# 4 ROBUST PLANNABLE REPRESENTATIONS

Enabling complex real robotics tasks requires extending current model-based methods to low-level high-dimensional observations. However, in order to do so, we need to specify which space they should operate on. Real-world environments are unstructured, cluttered, and present distractors. Our approach, MIRO, is able to learn latent representations that capture the relevant aspects of the dynamics and the task by framing the representational learning problem in information theoretic terms: maximizing the mutual information between the latent space and the future observations. This objective advocates for representing just the relevant aspects of the dynamics, removing the burden of reconstructing the entire pixel observation.

We first motivate the use of the mutual information objective as a representational learning objective for control, then we derive our objective that entangles dynamics and reward learning with the representational objective, and finally we instantiate this objective in a concrete model-based algorithm.

## 4.1 ROBUSTNESS OF LEARNED LATENT SPACES

Representations that ignore the presence of elements unrelated to the task at hand would allow us to apply successfully off-the-shelf reinforcement learning algorithms in real-world environments. In contrast, current representational learning approaches for control are based on reconstruction objectives. Here we show that these methods are not suitable for real-world tasks, since even in the presence of simple distractors, they completely fail to capture the important aspects of the task.

Concretely, several prior work learns representations by reconstructing the observation using a variational auto-encoder (VAE) (Kingma & Welling, 2013):

$$\max_{\theta,\phi} \mathbb{E}_{q_\theta(z|x)}[\log p_\phi(x|z)] - D_{\mathrm{KL}}(q_\theta(z|x)\|p(z))$$

here $p(z) \sim \mathcal{N}(0, 1)$, and $\theta$, $\phi$ are the parameters of the encoder and decoder, respectively. This objective captures the variation of the input in the latent space while the commonalities are stored in the weights of the auto-encoder. As a result, an unstructured and changing environment will be encoded in the latent space even when this variation is irrelevant to the control problem.

The mutual information objective presents itself as an alternative to learn representations. We argue that this objective is more suitable for learning robust representations. In the POMDP setup, we propose to maximize the mutual information between past observations and future observations, i.e., $\max_\theta I(o_{1:t}; o_{t+h}|a_{t:t+h})$, where $\theta$ are the parameters of the encoder (no decoder is needed). Then, given limited capacity of the encoder, the optimal representation should not capture the details on the scene that do not depend on the actions. Including more information in latent space always increases the mutual information, hence limiting the capacity or adding an information bottleneck is

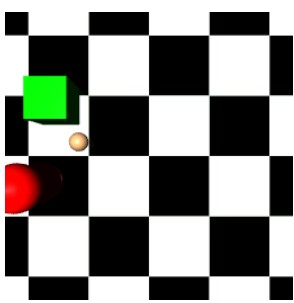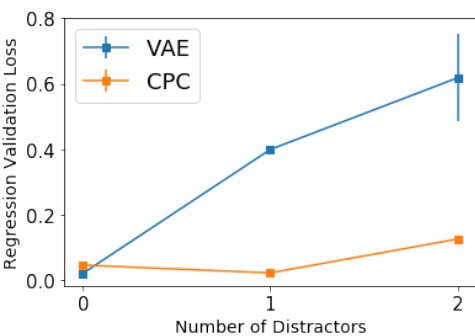

Figure 1: **Left.** Point-mass environment with distractors. **Right.** As a function of the number of the distractors, we plot the accuracy of the regression to the position of the point-mass when regressing from the latent representation learned by a variational autoencoder (VAE) and when regressing from the latent representation learned by contrastive predictive coding (CPC). Even in simple domains the reconstruction based objective for learning representations (VAE) discards the information needed for modelling the dynamics of the system, while the mutual information objective preserves it.

crucial for obtaining a robust representation. We estimate the mutual information using InfoNCE lower bound (van den Oord et al., 2018):

$$\max_\theta \mathbb{E}\left[\frac{f_\theta(o_{1:t}, o_{t+h})}{\sum_{h'} f_\theta(o_{1:t}, o_{t+h'})}\right]$$

Here, the expectation is on sets of observtions $\{o_{i_1}, ..., o_h, ..., o_{i_{N+1}}\}$ with N negative examples and one positive example, $o_h$, and $f_\theta$ represent a score function. The mutual information objective allows us to encode an inductive bias towards the important aspects of the task. The inductive bias arises from the fact that we are optimizing a discriminative model, instead of generative such as the VAE. Figure 1 shows the effect of distractors on the learned representation in a simple point-mass environment. Both representations are learned using the same data, raw pixel images of the point mass with the distractors. Then, we regress on top of the learned representations to the position of the point-mass. The mutual information objective allows to maintain the accuracy even when more distractors are present. However, the autoencoder accuracy is increasingly harmed by the appearance of unrelated objects to the task.

## 4.2 LEARNING LATENT SPACES FOR CONTROL

In this section, we develop the MIRO objective that entangles the representation, dynamics model, and reward predictor. Consequently, the objective not only learns a representation that is robust, but also emphasizes the important aspects of the dynamics and the task.

Figure 2 shows the POMDP set-up, where the only observed variables are the observations $o_t$ and rewards $r_t$, and we are trying to model the underlying latent space $s_t$ and the underlying process that generates the observed variables. We model the latent space $s_t$ and reward $r_t$ to be Gaussian with diagonal covariance given the previous state and action, i.e., $s_{t+1}|s_t, a_t \sim \mathcal{N}(\mu_{t+1}, \Sigma_{t+1})$ and $r_t|s_t, a_t \sim \mathcal{N}(\nu_t, \Lambda_t)$. In order to learn the underlying process of the POMDP and successfully use this representation for control, we learn four functions: an encoder, filter, transition dynamics, and reward predictor. In order to plan, we need to process the current observation (encoder), update the latent space with the current observation (filter), predict the

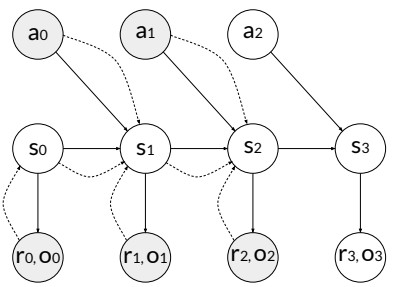

Figure 2: Probabilistic graphical model of the POMDP. The only variables observed (shaded nodes) are the actions, observations (high-dimensional images), and rewards. Our model has to infer the latent space (dashed lines) as well as to model the conditional probabilities (solid lines).

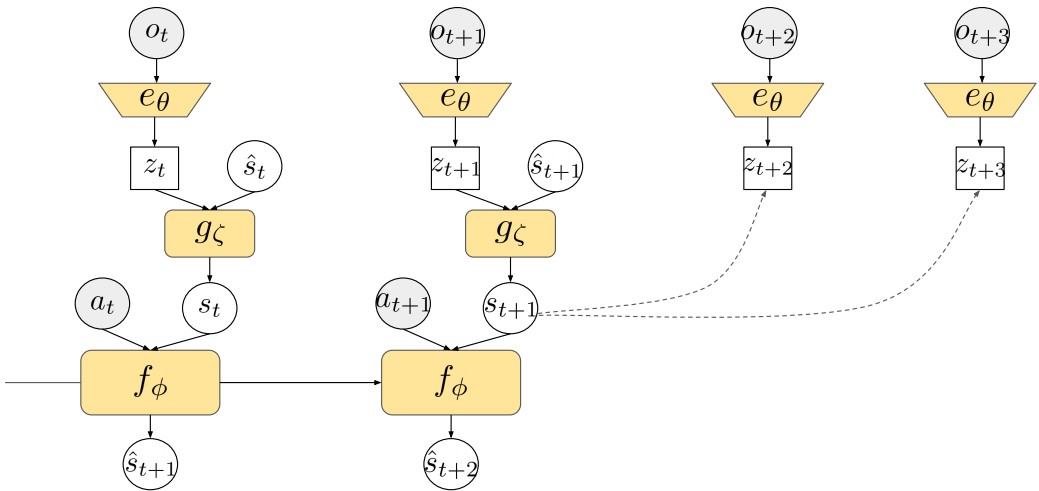

Figure 3: Proposed architecture that optimizes the MIRO objective. The shaded variables are observed, while the unshaded are latent, the yellow nodes represent parametric functions. The observation $o_t$ is fed into the encoder resulting in the intermediate latent variable $z_t$. Together with the prior on the latent state $\hat{s}_t$ constitute the input of the filter, modelled as a feed-forward neural network. The filtered state $s_t$ and action $a_t$ is passed through the dynamics model, which is a recurrent neural network, to obtain the predicted (prior) latent for the next state $\hat{s}_{t+1}$. The dashed lines denote the variables between which we maximize the mutual information. (Circle: stochastic; square: deterministic). Reward predictor network is omitted.

next states (transition dynamics) and predict the future rewards (reward predictor). In the following, we describe in detail the parametrization, and functionality of each of these components:

**Encoder.** The encoder parametrizes a map from a high-dimensional observation to a lower dimensional manifold, $e_\theta : \mathcal{O} \to \mathcal{Z} \subseteq \mathbb{R}^m$. We represent the encoded observation $e_\theta(o_t)$ with the notation $z_t \in \mathcal{Z}$. Contrary to prior work in learning latent spaces for planning, we do not make any assumption on the underlying distribution of $z_t$ or $o_t$ (such as they are Gaussian (Kingma & Welling, 2013)).

**Filter.** This function, $g_\zeta : \mathcal{Z} \times \mathcal{S} \to \mathcal{S}$, filters the belief of the prior $\hat{s}_t$ state variable with the current encoded observation $z_t$ resulting in the filtered variable $s_t$. Essentially, it updates the latent to be coherent with the most recent observation.

**Dynamics model.** The dynamics model approximates the transition function on the learned latent space, $f_\phi : \mathcal{S} \times \mathcal{A} \to \mathcal{S}$. The dynamics parametrizes the mean $\mu_t$ and diagonal covariance $\Sigma_t$ of the underlying transition distribution of the latent process. We denote by $\hat{s}_{t+1}$ the prior latent variable resulting from sampling from the dynamics model distribution.

**Reward predictor.** The reward predictor regresses onto the observed rewards from the latent variable. Hence, it is a mapping from latent states to rewards, $r_\psi : \mathcal{S} \to \mathbb{R}$. As said, we assume that the rewards follow a Gaussian distribution with unit variance.

We denote by $\Theta = \{\theta, \zeta, \phi, \psi\}$ the parameters of our function approximators. These are learned altogether with the following constrained optimization objective:

$$\max_{\Theta} \quad I(s_t; o_{t+h} | a_{t:t+h})$$
$$\text{s.t.:} \quad D_{\text{KL}}(g_\zeta(z_{t+1}, \hat{s}_{t+1}) \| s_{t+1}) \le \epsilon$$
$$D_{\text{KL}}(r_t \| r_\psi(s_t, a_t)) \le \epsilon$$

As seen in the previous section, the mutual information maximization term allows us to obtain latent spaces that just contain information about the relevant components of the scene. The enforcement of the constraints has two purposes: 1) ensures small prediction errors when planning, and 2) guides the state space to capture the components for predicting rewards and future states.

In practice, the previous objective is intractable to optimize. The exact maximization, or even evaluation, of the mutual information term is intractable, as well as the set of non-linear inequalities.

---

**Algorithm 1** Model-Based Reinforcement Learning with MIRO

---

1: Initialize encoder $e_\theta$, filter $g_\zeta$, dynamics model $f_\phi$, reward predictor $r_\psi$ and dataset $\mathcal{D} \leftarrow \emptyset$.
2: **repeat**
3:     $\mathcal{T} \leftarrow \emptyset$
4:     **for** $k = 1 \dots K$ **do**
5:         Sample initial state $s_0 \sim \rho_0$
6:         **for** $t = 0 \dots H$ **do**
7:             Filter the state $\hat{s}_t$ with $o_t$ using $g_\zeta$ to get $s_t$
8:             Get action $a_t$ from planner using $s_t$, $f_\phi$ and $r_\psi$
9:             Take action $a_t$ in the environment and obtain $o_{t+1}, r_t$
10:           $\mathcal{T} \leftarrow \mathcal{T} \cup \{s_t, a_t, r_t, o_t\}$
11:           Estimate next state $\hat{s}_{t+1} \sim f_\phi(s_t, a_t)$
12:         **end for**
13:         $\mathcal{D} \leftarrow \mathcal{D} \cup \mathcal{T}$
14:     **end for**
15:     Optimize Equation 1 with mini-batches from dataset $\mathcal{D}$
16: **until** Until desired performance is achieved
17: **return** Optimal model parameters $\Theta$

---

Instead, we optimize a tractable lower bound of this objective. First, we replace the mutual information term with the noise constrative estimator lower bound $I_\text{NCE}$. Second, we formulate its Lagrangian and treat the dual parameters as hyperparameters. Altogether, it gives an optimization objective that can be easily optimized with stochastic gradient descent

$$\max_\Theta I_\text{NCE}(s_t; o_{t+h}|a_{t:t+h}) - \lambda_1 D_\text{KL}(g_\zeta(z_t, \hat{s}_{t+1})\|s_{t+1}) - \lambda_2 D_\text{KL}(r_t\|r_\psi(s_t, a_t)) \tag{1}$$

We optimize this objective by using Monte-Carlo estimates and backpropagating through the stochastic nodes with the path-wise derivative, also known as reparametrization trick (Kingma & Welling, 2013). As in van den Oord et al. (2018), we use a log-bilinear model for the score function, which leads to the following formulation of the InfoNCE objective. Be $x_t = [s_t; a_{t:t+h}]$:

$$I_\text{NCE}(s_t; o_{t+h}|a_{t:t+h}) = \mathbb{E}\left[\log\left(\frac{\exp(x_t^\top W_h z_{t+h})}{\exp(x_t^\top W_h z_{t+h}) + \sum_{j=1}^N \exp(x_t^\top W_h z_j)}\right)\right] \tag{2}$$

Even though more complicated models can be used, as well as models that encode the future sequences of actions; This simple score model tends to work well in practice.

## 5   ROBUST LATENT SPACE MODEL-BASED REINFORCEMENT LEARNING

In this section, we present our architecture, implementation, and instantiation into a model-based method. The presented algorithm uses planning, but our representational learning method for control is agnostic to the derived controller. Methods such as Lee et al. (2019) could be used with MIRO. The pseudocode for the model-based approach is shown in Alg. 1, and is typical in model-based RL algorithms, it iterates between three steps: 1) data collection, 2) model learning, and 3) policy improvement or planning.

**Data Collection.** At each iteration, we first collect data by planning with the current dynamics model. On-policy data collection relieves the model from learning the entire space, and instead it just focuses on the regions that the agents visits. It also allows to overcome the insufficient coverage of the initial data distribution. The data collected is stored in a replay buffer that is used for training the model.

**Model Learning.** The architecture of the model is shown in Figure 3. It is composed by an convolutional encoder, a feed forward filter, and a dynamics model that has the form of a recurrent neural network. The reward predictor, not shown in Figure 3, is modelled as a feed forward neural network that has as input the latent state variable. The model is trained using all the data collected so far on the MIRO objective (Equation 1).

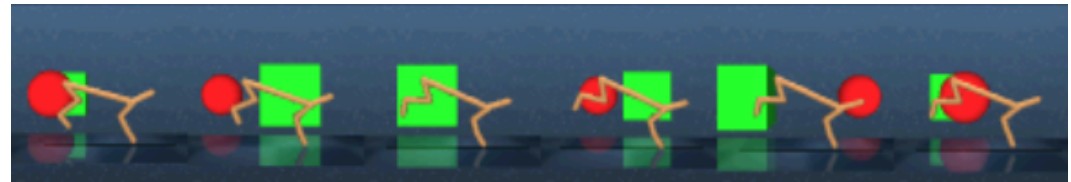

Figure 4: An example trajectory of Cheetah environment with distractors. The distractor objects (red sphere and green cube) are placed at a random position at each time step.

**Planning.** In this work, we use model-predictive control (MPC) with cross-entropy method (CEM) for action selection (Botev et al., 2013). The CEM algorithm selects the action sequence that maximizes the expected return under the learned dynamics. Specifically, it is a population based procedure that iteratively refits a Gaussian distribution, starting from a unit Gaussian, on the best sequences of actions. The MPC component prevents overfitting to the model deficiencies by selecting the best CEM action and replanning at each step. Note that our represenational learning objective is orthogonal to the planning method used.

# 6 RESULTS

In this section, we empirically corroborate the claims in the previous sections. Specifically, the experiments are designed to address the following questions:

1. Is our approach able to maintain its performance in front of distractors in the scene?
2. How does our method compare with state-of-the-art reconstruction objectives?
3. Does our latent variable remain unchanged in the presence of visual noise?

To answer the posed questions, we evaluate our framework, in four continuous control benchmark tasks MuJoCo simulator: cartpole-balance, reacher, finger-spin and half-cheetah (Todorov et al., 2012; Tassa et al., 2018). We choose PlaNet (Hafner et al., 2018) as the state-of-the-art reconstruction objective baseline for plannable representations.

## 6.1 ROBUST CONTINUOUS CONTROL COMPARISON

To test the robustness of MIRO and PlaNet in visually noisy environments, we add distractors to each of the four environments, as shown in Fig. 4.

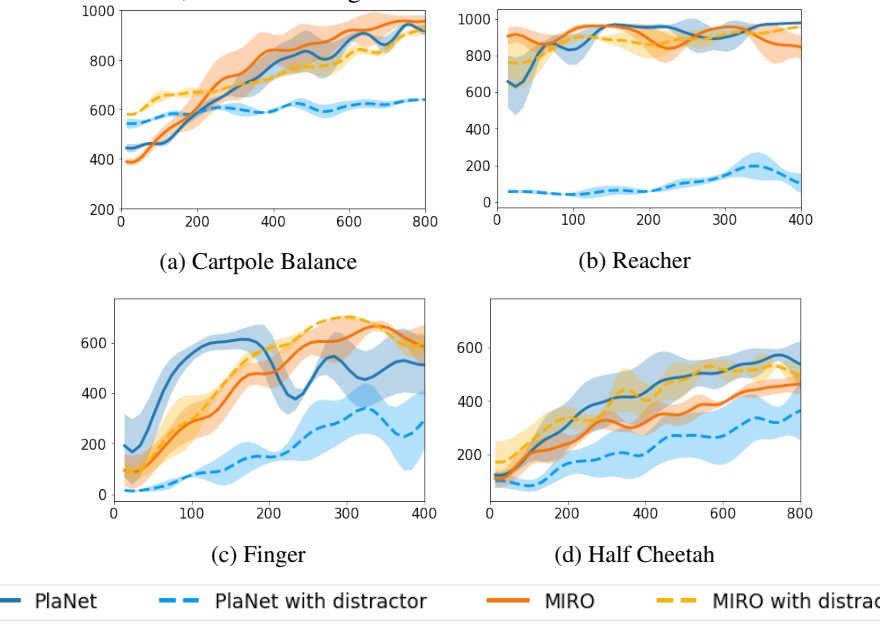

Figure 5: Learning curves of MIRO and PlaNet on environments with and without distractors. y-axis plots the cumulative reward while x-axis plots number of rollouts collected. All curves represent mean and the shaded area represents one standard deviation among 3 seeds. MIRO's performance remains unchange in the presence of disctractor objects in the environments while PlaNet is drastically affected by such objects. When no distractors are present in the scene both approaches have comparable performance.

The results, depicted in Figure 5, show that in all four environments the performance of MIRO is not undermined by the presence of objects unrelated to the task. Interestingly, in the half-cheetah environment, the performance even improves with visual noise in the background. We argue that the presence of visual noise forces the embedding to focus even more on information relevant to the task, and thus makes the embedding more suitable for planning. In comparison, the performance of PlaNet struggles in face of distractors: the agent is unable to learn in the reacher environment, and struggle to learn in the other environment obtaining sub-optimal performance. When no distractors are present in the scene, Figure 5, shows that our approach achieves comparable or superior performance than its reconstruction-based counterpart.

## 6.2 REALISTIC DISTRACTORS

In the previous section, the distractors added to the scene is placed at a random position at each time step. While this resembles particular types of random noise, the movement of majority of objects in real life do not conform to such behavior. To simulate a more realistic visual noise, we add the same geometric objects in 4 to the scene but move the $i$-th distractor according to the following linear rule:

$$(x_t^i + 1, y_t^i + 1) = (x_t^i, y_t^i) + l\tilde{\mathbf{u}}^i$$

where $l$ is size of each step, $\tilde{\mathbf{u}}^i$ is a unit vector denoting the direction of the movement and it is resampled with probability $p = 0.2$ at each timestep. We term the resulting distractors "linear distractors" and term the distractors Figure 4 "random distractors".

We compared two ways of obtaining a dynamics model on tasks with linear distractors: 1. **Transfer**: Train with random distractors and test on linear distractors; 2. **From scratch**: Train with linear distractors and test on linear distractors. The performance of both MIRO and PlaNet on two schemes are presented in table. The results showcase that MIRO, when presented with realistic distractors, is still able to perform the task. Since MIRO is maximizing an MI objective, the latent space is more robust to the distractors when it is exposed to higher entropy distractors at training time.

|                   | PlaNet (Transfer) | MIRO (Transfer) | PlaNet (From scratch) | MIRO (From scratch) |
|-------------------|-------------------|-----------------|-----------------------|---------------------|
| Cartpole Balance  | $615 \pm 31$      | $984 \pm 6$     | $583 \pm 57$          | $780 \pm 71$        |
| Reacher           | $139 \pm 126$     | $673 \pm 148$   | $274 \pm 238$         | $551 \pm 43$        |
| Finger            | $512 \pm 21$      | $784 \pm 17$    | $582 \pm 31$          | $648 \pm 50$        |
| Half Cheetah      | $308 \pm 128$     | $615 \pm 62$    | $211 \pm 73$          | $576 \pm 148$       |

## 6.3 UNDERSTANDING THE LEARNED LATENT SPACE

Here, we provide insight into robustness of latent learned by MIRO. An ideal latent space should only encode information essential to the dynamics of the task. Thus, given observations $o_{\text{clean}}$ and $o_{\text{dis}}$, the later being the same as the former but with added distractors in the background, the desired behaviour of our encoder would be to output the same latent varible; i.e., $z_{\text{clean}} \approx z_{\text{dis}}$, being $z_{\text{clean}}$ the encoded $o_{\text{clean}}$ ($e_\theta(o_{\text{clean}})$) and $z_{\text{dis}}$ the encoded $o_{\text{dis}}$ ($e_\theta(o_{\text{clean}})$). In order to quantify the discrepancy on the latent, we measure its normalized difference $\frac{\|z_{\text{clean}} - z_{\text{dis}}\|_2}{\|z_{\text{clean}}\|_2}$. Figure 6 shows the average normalized difference on the Cartpole Balance environment when the latent space is learned using the MIRO objective and a sequential VAE objective (Hafner et al., 2018; Lee et al., 2019; Ha & Schmidhuber, 2018). The results manifest that the embedding learned by MIRO excludes irrelevant information over the course of training. In

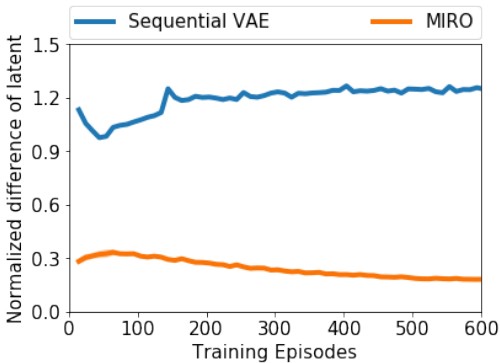

Figure 6: Normalized different in the latent variable when encoding the same observation with and without distractors. MIRO learns representations that are invariant to distractors, resulting in lower difference, while the embedding learned by the sequential VAE fails to capture the relevant aspects of the dynamics despite being trained jointly with the dynamics.

contrast, with sequential VAE, the normalized error increases and lands at higher final value, which indicates that the model fails to understand the essential elements in the task despite being trained jointly with the dynamics.

## 7 CONCLUSION

In this paper, we present MIRO, an information theoretic representational learning objective that tackles the problem of learning robust and generalizable representations for control. Our approach maximizes the mutual information between the learned latent space and future observations, emphasizing the features that are relevant for the dynamics and the reward predictions. We compare our approach in standard benchmark control environments with and without visual disturbances. Our method remains unaffected by those disturbances while state-of-the-art methods fail to complete the tasks. When disturbances are removed it attains comparable or superior performance to prior work. Furthermore, we experimentally investigate the effect of such disturbances in the latent variables, showing that MIRO's latents are invariant to visual noises. The development of this method has been motivated for the applicability of general reinforcement learning to real-world and unstructured environments. Consequently, an enticing direction for future work would be to analyze MIRO's behaviour in real-robotic agents outside structured environments.

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

## 8 APPENDIX

### 8.1 HYPERPARAMETERS

**Environment:** Dimension of the input observation is $32 \times 32 \times 3$. Horizon of all the tasks are $H = 1000$ with the same action repeat parameters as in (Hafner et al., 2018).

**Network:** The dimension of the observation encoding space $\mathcal{Z}$ is 1024, dimension of the latent space $\mathcal{S}$ is 30. Horizon of the CEM planner is 12. The horizon $h$ of the Mutual Information Objective is 3. Number of negative examples for NCE objective is $N = 10$.

**Loss Function:** In equation 1, the coefficient for dynamics model loss is set to $\lambda_1 = 0.002$, the coefficient for reward prediction loss is set to $\lambda_2 = 0.02$.

### 8.2 ABLATION STUDIES

In the following we present the ablation studies.

#### 8.2.1 MUTUAL INFORMATION OBJECTIVE

To study the effect of the mutual information objective in the loss function, we ablated the MI objective in 1 and compared its performance with our full model (MIRO). As shown in 7, using solely dynamics model loss and reward prediction loss, planning fails on all four environments. This shows that the MI objective is a central part of the algorithm.

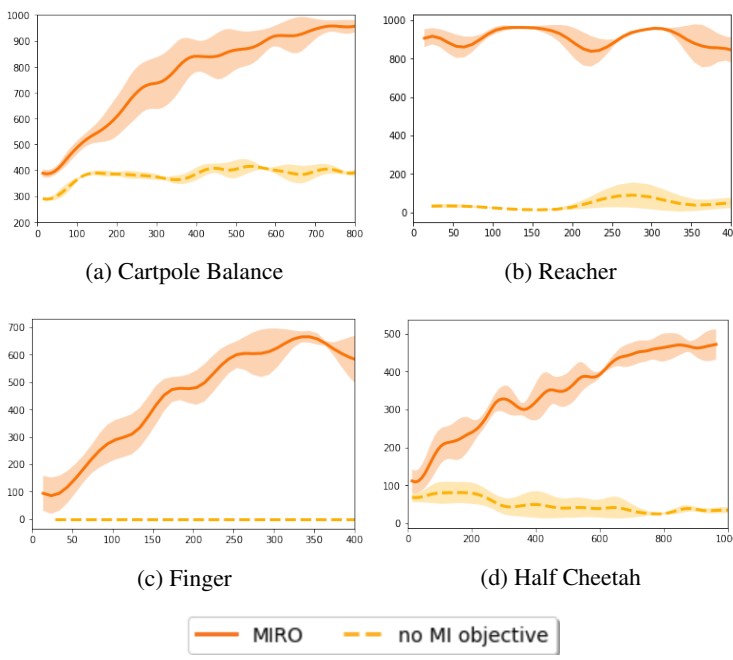

Figure 7: Learning curve of MIRO and an ablated version without MI objective.

#### 8.2.2 MODEL ARCHITECTURE

In this section we study the effect two architecture choices of the recurrent neural network used in the model. We compare the performance when the latent state is stochastic (as used in MIRO) and when the latent state is deterministic as shown in Figure 9.

With deterministic latent space, the performance hardly improves. This could be due to the fact that deterministic latents are more prone to overfit. The observation is consistent with (Hafner et al., 2018).

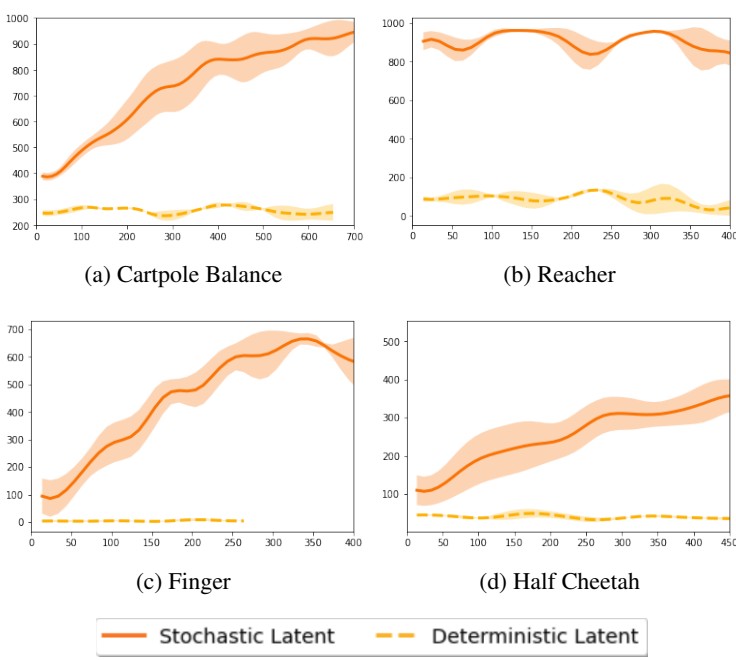

Figure 8: Comparison of deterministic latent space and stochastic latent space.

### 8.2.3 Mutual Information Conditioned on Actions

In this section we compare two different ways of incorporating actions in the mutual information objective. When predicting $z_{t+k}$ from $s_t$, we can incorporate actions $a_t, \cdots, a_{t+k-1}$ in two ways: 1. stacking the actions and concatenating them with $s_t$ when predicting $z_{t+k}$; 2. execute an open-loop rollout from $s_t$ given $a_t, \cdots, a_{t+k-1}$ and get $\hat{s}_{t+k}$, and predict $z_{t+k}$ from $\hat{s}_{t+k}$. For the first variant, we also examined the effect of different horizon $h$.

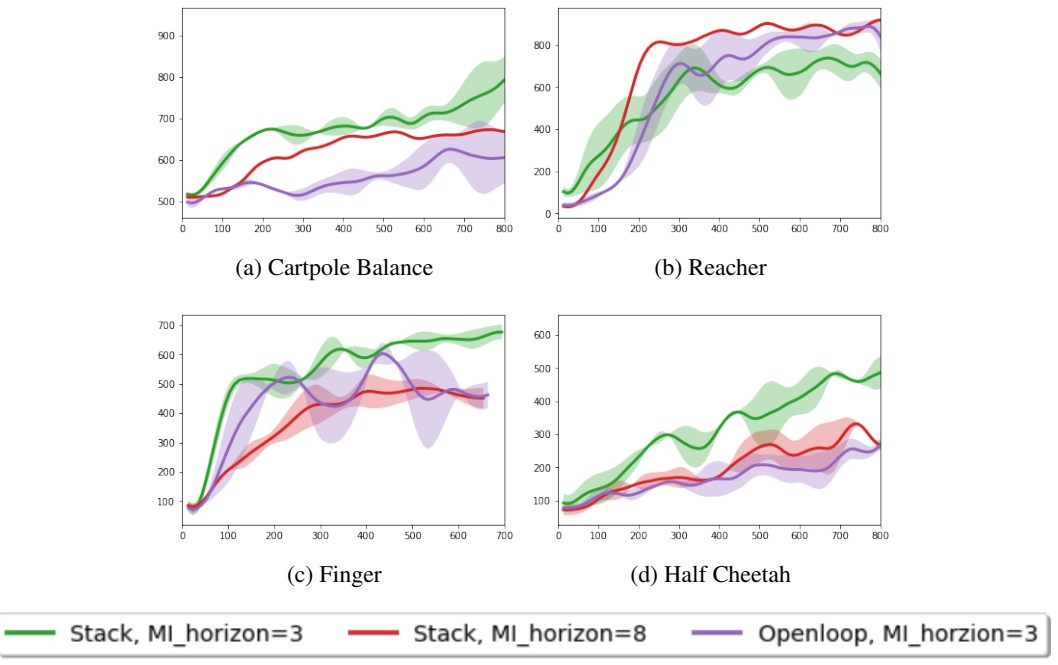

Figure 9: Different ways of conditioning the context vector on actions.

