# OpenReview forum: "Mutual Information Maximization for Robust Plannable Representations"
_ICLR.cc/2020/Conference — Reject_

### Official Review · AnonReviewer2 · 2019-10-23
**Official Blind Review #2**

**Rating:** 1

**Review:**

The authors propose a model-based reinforcement algorithm in which the model is a sequential latent variable model and the actions planned with a cross-entropy method (CEM) planner. The model is learnt by maximizing a lower bound on the mutual information between the latent states and their successor observations (instead of the classical sequential ELBO). The authors argue that the latter objective function yield robustness to distraction in visual scenes. The algorithm, named MIRO, is experimented on 4 simulated environments.
---
Overall I did not find the paper particularly clear and easy to read. The method is only introduced in the 5th page and no ablation study is conducted.
It is still not obvious to me why maximizing the MI in the objective function would reduce the influence of potential distractors.`
Furthermore, the paper overlooks a good part of the related work on extending VAEs to sequence data, published in the last 3 years and does not draw links to similar architectures.
The experiments are in my opinion not convincing, as the approach is only experimented on 2 non trivial -yet not particularly challenging- environments (Finger and Half Cheetah).

Minor: in the equation of the ELBO, page 3, the parameters \theta and \phi are swapped.


**Experience Assessment:**

I have read many papers in this area.

**Review Assessment: Checking Correctness Of Derivations And Theory:**

I assessed the sensibility of the derivations and theory.

**Review Assessment: Checking Correctness Of Experiments:**

I carefully checked the experiments.

**Review Assessment: Thoroughness In Paper Reading:**

I read the paper at least twice and used my best judgement in assessing the paper.

---

> ### Author Response · Authors · 2019-11-15
> **Response to Reviewer 2**
>
> We thank the reviewer for their thoughtful and constructive comments. As suggested by the reviewers, we ran additional experiments and clarified sections of the paper. The additional experiments and modifications are: 1) Added a section with realistic distractors, 2) run several ablations, such as architectural choices, ablating the mutual information in the objective, and sensitive to different prediction horizon; 3) extended the related work section, and 4) added the hyperparameters and network configuration.
>
> >The paper overlooks a good part of the related work on extending VAEs to sequence data
> We have extended the related work section to include the previous work mentioned by reviewer 1, and more references regarding sequential VAEs.
> >The method is only introduced in the 5th page
> We motivate one of the key components of our method on page 3. The method is introduced in page 4 and 5.
>
> > No ablation study is conducted
> An appendix has been added to the paper that contains an ablation study of different architecture choices: deterministic versus probabilistic latent space, choice of incorporating the actions into the CPC objective, and the removal of the mutual information objective. We also evaluate the sensitivity of our method to the CPC horizon. This ablation study underpins the choice of our architecture.
>
> >The experiments are in my opinion not convincing
> Approaches that use shooting with model-predictive control, while effective in low-dimensional domains they do not scale up well to more complex domains. Our evaluated domains are the ones commonly tested when planning with such methods from state-space[1], and images [2].
>
> We hope that with these clarifications and further analysis of our method the reviewer will consider our work for acceptance.
>
>
> [1] Deep Reinforcement Learning in a Handful of Trials using Probabilistic Dynamics Models. Kurtland Chua, Roberto Calandra, Rowan McAllister, Sergey Levine
> [2] Learning Latent Dynamics for Planning from Pixels. Danijar Hafner, Timothy Lillicrap, Ian Fischer, Ruben Villegas, David Ha, Honglak Lee, James Davidson.

---

### Official Review · AnonReviewer3 · 2019-10-23
**Official Blind Review #3**

**Rating:** 6

**Review:**

The authors propose a latent dynamics model that is learned by maximizing a bound on mutual information between image embeddings and the latent state h time steps later. The model is evaluated on four standard visual control tasks that are solved by online planning.

Strengths:
- The paper addresses an important open problem with latent dynamics models.
- The paper is written clearly.

Weaknesses:
- The paper does not discuss why the mutual information is expected to pay less attention to distractor objects.
- The paper repeatedly mentions "task relevant" information. However, I cannot find anything about the method that would make the learned features more task relevant than reconstruction. This should be clarified.
- The distractor objects in the experiments randomly change locations in each frame. How would the model be expected to behave if they changed in a predictable way?
- The paper is lacking detail about hyper parameters and model architecture. What value for h is used? Is the transition function a vanilla RNN?
- The paper is missing an ablation study. It would be interesting which of the design choices about the model contribute to its success.

Questions:
- Could you please explain the KL term that is weighted by lambda_2 in Eq 1? The KL notation on random variables rather than distributions seems non-standard. It is unclear why information about the reward should be penalized.
- Is the objective summed across time steps? How is the data sampled that the model trains on?

Comments:
- The paper claims in the introduction that reconstruction based approaches cannot discard low-level information. This claim should be rephrased, since the decoder variance allows to discard low-level information (the amount can be controlled by scaling the KL).
- I would suggest to remove the word "robust" from the title or find a more descriptive term to replace it with.

**Experience Assessment:**

I have published in this field for several years.

**Review Assessment: Checking Correctness Of Derivations And Theory:**

I carefully checked the derivations and theory.

**Review Assessment: Checking Correctness Of Experiments:**

I carefully checked the experiments.

**Review Assessment: Thoroughness In Paper Reading:**

I read the paper thoroughly.

---

> ### Author Response · Authors · 2019-11-15
> **Response to Reviewer 3**
>
> We thank the reviewer for their thoughtful and constructive comments. As suggested by the reviewers, we ran additional experiments and clarified sections of the paper. The additional experiments and modifications are: 1) Added a section with realistic distractors, 2) run several ablations, such as architectural choices, ablating the mutual information in the objective, and sensitive to different prediction horizon; 3) extended the related work section, and 4) added the hyperparameters and network configuration.
>
> In the following, we address the specifics of the reviewer. First, the weaknesses; second, the question; and lastly, the comments.
> Weaknesses:
> 1. It was our aim to motivate this in section 4.1. We have clarified the text and discussed why the mutual information objective is expected to pay less attention to distractors. The intuition is: although encoding more information in the latent space increases mutual information across time steps, when the latent space has limited capacity, it is incentivized to pivot to parts of the state space that contributes most information gain and disregard the elements that brings little information gain (distractors).
> 2. Our assumption is that task relevant information are elements in the scene that can be altered by agent actions.
> 3. We added a new section 6.2 and extended our evaluation to more realistic and predictable distractors that follow a coherent motion. We investigated two schemes: 1) train on distractors that randomly chooses positions between frames and test on predictable distractors, and 2) train and test on predictable distractors. The results show that our approach successfully performs the tasks when predictable distractors are present, while (1) having better performance than (2). This result is not surprising: since we are maximizing the mutual information, our learned latent space will ignore components of the image with high entropy.
> 4. We have added on the appendix a hyperparameter section that should contain all the details of the architecture. Regarding the specific questions of the reviewer, the horizon for NCE objective is h=3. We used a vanilla RNN with a probabilistic latent space.
> 5. In the appendix, we have also incorporated an ablation study that studies architectural choices, variation of loss functions, and sensitivity to prediction horizon.
>
> Questions:
> 1. The information theoretic notation on random variables is used in the standard literature (for instance see [1]). Since we model the reward as a Gaussian variable, we try to minimize the KL divergence between in our reward predictor and the true reward in order to train an accurate reward predictor. In practice, this amounts to train the reward predictor by maximum likelihood.
> 2. The objective is summed across time-steps. One new trajectory is sampled every 5000 gradient steps taken. The actions are determined by CEM planner based on the current model (plus exploratory noise) . We have modified the paper to reflect it.
>
> Comments:
> 1. We fully agree with the reviewer, and we have removed the claim in the introduction.
> 2. We will modify the title to be more descriptive.
>
> We hope that with these clarifications and further analysis of our method the reviewer will consider our work for acceptance.
>
>
> [1]  Elements of Information Theory. Thomas M. Cover and Joy A. Thomas.

---

### Official Review · AnonReviewer1 · 2019-10-24
**Official Blind Review #1**

**Rating:** 3

**Review:**


########### Post-rebuttal summary ############
The proposed method relies on the fact that the distractors have highly unpredictable movements such that a mutual information objective between frames of a sequence learns to ignore them. The experimental evaluations are performed with distractors that randomly change positions / directions, which fits the requirements of the method but not necessarily the behavior of distractors in real environments. To properly evaluate this, experiments in environments with more realistic distractors (who's dynamics are not chosen by the authors) are necessary. Therefore, I do not recommend acceptance of this submission.
########################################


## Paper Summary

This paper combines a mutual information maximisation objective a la CPC with objectives for dynamics and reward prediction to learn a representation for downstream planning / skill learning that is more robust to visual distractors than comparable representations learned with reconstruction-based objectives a la VAE. In particular, the MI objective maximises the mutual information between the representation of the current state and future observations. The authors show improved robustness of their representation to visual distractors over a baseline with pixel-reconstruction-based representation learning (PlaNet). As a result they are able to achieve better performance when model-predictive control is used on top fo the learned representation to perform control on simulated DM Control Suite tasks with added simple visual distractors.


## Strengths

- the paper addresses a relevant problem with an intuitive approach

- the paper is well written and easy to read

- the analytic toy experiments at the end of Sec 4.2 and in 6.2 help understand the properties of the learned representation

- the proposed method, when applied to the model-based algorithm PlaNet, shows improved robustness in settings with artificially introduced visual distractors in simulated DM Control Suite tasks


## Weaknesses

(1) unclear whether artificial distractors are indicative of behavior with real distractors: the fact that distractors do not follow coherent motions but instead randomly change position between two consecutive frames makes it hard to estimate how this would perform on more natural distractors. As there is no MI between the distractor's position in one image and any other image, a CPC-style representation learning objective will naturally be encouraged to ignore them. However, if they move with more natural, coherent motion this might not be the case. I would suggest to add an experiment with more natural distractor motion (see below).

(2) it is unclear how much of the invariance to distractors is coming from the MI objective and how much simply from the fact that the representation is learned with jointly predicting the reward. In order to justify that the MI objective is helpful for learning the representation I would suggest to run an ablation experiment that trains using *only* the reward prediction objective and compare performance. (see also suggested experiment below)

(3) parts of the model formulation require clarification: when reading Sec 4.2 that describes the model some parts were unclear to me (see concrete questions below).

(4) the amount of detail provided in the paper is insufficient for reproducing the results: the paper lacks detailed information about the used architectures, hyperparameters and versions of the baselines employed (e.g. PlaNet with stochastic/deterministic prediction?), code is not provided.

(5) lacks reference to recent work on CPC-style objectives for RL (see suggested references below)


## Questions

(A) why is the latent state variable s_t observed in the model depicted in Fig 3 (i.e. part of the input data)? Shouldn't this variable be latent? How can the KL constraint on it be computed if it is not observed in the input data?
(B) should the first term on the right hand side of equation (2) have \hat{s}_t instead of \hat{s}_{t+h}? Otherwise, how is the I_{NCE} computation conditioned on the current state s_t?
(C) what is the scale of the x-axis in Fig 5, i.e. does it show the number of environment rollouts / steps or the number of model re-training iterations? if the latter is the case, how does that translate to the number of environment interactions?


## Suggestions to improve the paper

(for 1) add an experiment where the distractor has more natural dynamics so that there is MI between the distractor positions in consecutive frames (e.g. ball bouncing in the image frame in the background instead of randomly jumping to new positions)
(for 2) add an experiment with a reward-prediction only baseline, i.e. only action-conditioned reward prediction so that task-irrelevant parts are ignored by default (i.e. also no reconstruction objective, but also no MI objective) -> show how PlaNet performance compares to the so far reported numbers when using this representation for planning
(for 4) add details about the architecture, hyperparameters and training schedule, for both the method and all comparisons to the appendix
(for 5) add references to related works that use CPC-style MI objectives for representation learning in the context of RL/skill learning:
	- [1] Nachum et al., ICLR 2019 -- applies CPC-style objective to hierarchical RL setting
	- [2] Anand et al., NeurIPS 2019 -- investigates MI objectives for representation learning on a wide range of Atari games (don't apply to RL)
	- [3] Gregor et al., NeurIPS 2019 -- while the main proposed model is generative they compare to a contrastive version that uses CPC to learn predictive representations (don't use it for RL)
	- [4] Guo et al., Arxiv 2018 -- similar investigation to [3] of CPC-style objective for representation learning in RL environments (don't use it for RL)
	- it should also be mentioned that the original CPC paper already showed that adding CPC-style auxiliary loss to RL improves performance (even though they did not compare to other model-based methods)
- add qualitative rollouts for predictions from the PlaNet predictive network both with and without distractor to the appendix


## Minor Edit Suggestions
- "Learning latent dynamics from pixels", Hafner et al. is cited twice in the reference section
- it might help to add the reward prediction module to Fig 3 or mention in the caption that it is omitted, it is only described later in the text and was confusing for me on first sight


[Novelty]: okay
[technical novelty]: minor
[Experimental Design]: okay
[potential impact]: high


#######################
[overall recommendation]: weakReject - I am inclined to accept this paper but am not fully convinced that the random distractors provide a good intuition about how the proposed method would behave with more natural distractors. If the authors are able to report positive results on the two requested experiments I am willing to raise my score.
[Confidence]: High


[1] Near-Optimal Representation Learning for Hierarchical Reinforcement Learning, Nachum et al., ICLR 2019
[2] Unsupervised State Representation Learning in Atari, Anand et al., NeurIPS 2019
[3] Shaping Belief States with Generative Environment Models for RL, Gregor et al., NeurIPS 2019
[4] Neural Predictive Belief Representations, Guo et al., Arxiv 2018



### Rebuttal Comment (copied here so that it's visible to the authors) ###

I appreciate the effort the reviewers put into the rebuttal! My main concern was that the mutual information objective only encourages the model to ignore the distractors because they change positions randomly between frames, i.e. there is no mutual information between distractor positions in consecutive frames. On the other hand, if the distractor's motion was perfectly deterministic there would be infinite mutual information between distractor positions in consecutive frames and therefore the representation might exclusively model the distractor. I.e. the more predictable the distractor, the worse the proposed method will perform.

I asked the authors to perform an experiment with a more predictable distractor that bounces in the image frame to test this hypothesis. The authors instead chose a distractor that moves predictably for a few steps before randomly changing movement directions. While this sounds similar it can actually make a big difference depending on how frames are sampled for the CPC objective, i.e. if the required pair of frames is sampled across a random direction change there is again no mutual information between the distractor positions in both frames.

In real scenes the behavior of distractors likely lies somewhere in between these extremes: they will likely not be fully deterministic but certainly not have frequent moments of purely random direction changes. Therefore, to properly evaluate the merit of the proposed approach, experiments on more realistic / previously published environments are needed where the distractor dynamics are given and cannot be altered to better fit the proposed method.

Due to this fundamental concern about the method I cannot recommend acceptance of the submission.


**Experience Assessment:**

I have published one or two papers in this area.

**Review Assessment: Checking Correctness Of Derivations And Theory:**

I carefully checked the derivations and theory.

**Review Assessment: Checking Correctness Of Experiments:**

I carefully checked the experiments.

**Review Assessment: Thoroughness In Paper Reading:**

I read the paper thoroughly.

---

> ### Author Response · Authors · 2019-11-15
> **Response to Reviewer 1**
>
> We thank the reviewer for their thoughtful and constructive comments. As suggested by the reviewers, we ran additional experiments and clarified sections of the paper. The additional experiments and modifications are: 1) Added a section with realistic distractors, 2) run several ablations, such as architectural choices, ablating the mutual information in the objective, and sensitivity to different prediction horizon; 3) extended the related work section, and 4) added the hyperparameters and network configuration.
>
> In the following, we address the specifics of the reviewer. Specifically, we aimed to address all the suggestions that the reviewer pointed.
> 1. We added a new section 6.2 and extended our evaluation to more realistic and predictable distractors that follow a coherent motion. We investigated two schemes: 1) train on distractors that randomly chooses positions between frames and test on predictable distractors, and 2) train and test on predictable distractors. The results show that our approach successfully performs the tasks when predictable distractors are present, while (1) having better performance than (2). This result is not surprising: since we are maximizing the mutual information, our learned latent space will ignore components of the image with high entropy.
> 2. We added in the appendix an ablation study that, among other ablations, evaluates our method when just learning the reward prediction. This baseline completely fails at performing the task due to the weak signal that the reward provides.
> 3. We have modified section 4.2 to clarify the questions and improved readability, and the fixes for the reviewer questions:
> 3.A. It should be indeed a latent, and thus not shaded. Since we parameterize $s_t$ with a Gaussian distribution, the KL divergence can be calculated with closed form.
> 3.B. Yes, as depicted in Figure 3. It should be $s_t$ and not $\hat{s}_{t+k}$.
> 3.C. The scale is number of rollouts. The horizon of all the tasks are H=1000, so each rollout corresponds to 1000 environment interactions.
> 4. We have added on the appendix a hyperparameter section that should contain all the  details of the architecture.
> 5. We have extended the related work section to include the mentioned citations and other relevant related work.
>
> We hope that with these clarifications and further analysis of our method the reviewer will consider our work for acceptance.

---

### Decision · Program_Chairs · 2019-12-19

**Decision:**

Reject

**Comment:**

The manuscript concerns a mutual information maximization objective for dynamics model learning, with the aim of using this representation for planning / skill learning. The central claim is that this objective promotes robustness to visual distractors, compared with reconstruction-based objectives. The proposed method is evaluated on DeepMind Control Suite tasks from rendered pixel observations, modified to include simple visual distractors.

Reviewers concurred that the problem under consideration is important, and (for the most part) that the presentation was clear, though one reviewer disagreed, remarking that the method is only introduced on the 5th page. A central sticking point was whether the method would reliably give rise to representations that ignore distractors and preferentially encode task information. (I would note that a very similar phenomenon to the behaviour they describe has been empirically demonstrated before in Warde-Farley et al 2018, also on DM Control Suite tasks, where the most predictable/controllable elements of a scene are reliably imitated by a goal-conditioned policy trained against a MI-based reward). The distractors evaluated were criticized as unrealistically stochastic, that fully deterministic distractors may confound the procedure; while a revised version of the manuscript experimented with *less* random distractors, these distractors were still unpredictable at the scale of more than a few frames.

While the manuscript has improved considerably in several ways based on reviewer feedback, reviewers remain unconvinced by the empirical investigation, particularly the choice of distractors. I therefore recommend rejection at this time, while encouraging the authors to incorporate criticisms to strengthen a resubmission.